# Scoring epidemiological forecasts on transformed scales

**Nikos I. Bosse**[1,2,3]*, **Sam Abbott**[1,2], **Anne Cori**[4], **Edwin van Leeuwen**[1,3,5],
**Johannes Bracher**[6,7☯], **Sebastian Funk**[1,2,3☯]

**1** Department of Infectious Disease Epidemiology, London School of Hygiene & Tropical Medicine, London, United Kingdom, **2** Centre for the Mathematical Modelling of Infectious Diseases, London, United Kingdom, **3** NIHR Health Protection Research Unit in Modelling & Health Economics, **4** MRC Centre for Outbreak Analysis and Modelling, Department of Infectious Disease Epidemiology, School of Public Health, Imperial College London, London, United Kingdom, **5** Modelling & Economics Unit and NIHR Health Protection Research Unit in Modelling & Health Economics, UK Health Security Agency, London, United Kingdom, **6** Chair of Statistical Methods and Econometrics, Karlsruhe Institute of Technology, Karlsruhe, Germany, **7** Computational Statistics Group, Heidelberg Institute for Theoretical Studies, Heidelberg, Germany

☯ These authors contributed equally to this work.
* nikos.bosse@lshtm.ac.uk

**Data Availability Statement:** All code and data is available at https://github.com/epiforecasts/transformation-forecast-evaluation.

**Funding:** NIB received funding from the National Institute for Health and Care Research (NIHR)

## Abstract

Forecast evaluation is essential for the development of predictive epidemic models and can inform their use for public health decision-making. Common scores to evaluate epidemiological forecasts are the Continuous Ranked Probability Score (CRPS) and the Weighted Interval Score (WIS), which can be seen as measures of the absolute distance between the forecast distribution and the observation. However, applying these scores directly to predicted and observed incidence counts may not be the most appropriate due to the exponential nature of epidemic processes and the varying magnitudes of observed values across space and time. In this paper, we argue that transforming counts before applying scores such as the CRPS or WIS can effectively mitigate these difficulties and yield epidemiologically meaningful and easily interpretable results. Using the CRPS on log-transformed values as an example, we list three attractive properties: Firstly, it can be interpreted as a probabilistic version of a relative error. Secondly, it reflects how well models predicted the time-varying epidemic growth rate. And lastly, using arguments on variance-stabilizing transformations, it can be shown that under the assumption of a quadratic mean-variance relationship, the logarithmic transformation leads to expected CRPS values which are independent of the order of magnitude of the predicted quantity. Applying a transformation of $\log(x + 1)$ to data and forecasts from the European COVID-19 Forecast Hub, we find that it changes model rankings regardless of stratification by forecast date, location or target types. Situations in which models missed the beginning of upward swings are more strongly emphasised while failing to predict a downturn following a peak is less severely penalised when scoring transformed forecasts as opposed to untransformed ones. We conclude that appropriate transformations, of which the natural logarithm is only one particularly attractive option, should be considered when assessing the performance of different models in the context of infectious disease incidence.

Health Protection Research Unit (HPRU) in Modelling and Health Economics (grant code NIHR200908). SA's work was funded by the Wellcome Trust (grant: 210758/Z/18/Z). AC acknowledges funding from the MRC Centre for Global Infectious Disease Analysis (reference MR/R015600/1) jointly funded by the UK Medical Research Council (MRC) and the UK Foreign, Commonwealth & Development Office (FCDO), under the MRC/FCDO Concordat agreement and is also part of the EDCTP2 programme supported by the European Union; the Academy of Medical Sciences Springboard, funded by the Academy of Medical Sciences, Wellcome Trust, the Department for Business, Energy and Industrial Strategy, the British Heart Foundation, and Diabetes UK (reference SBF005\1044); and the National Institute for Health and Care Research (NIHR) Health Protection Research Unit in Modelling and Health Economics, a partnership between the UK Health Security Agency, Imperial College London and LSHTM (grant code NIHR200908). EvL acknowledges funding by the National Institute for Health and Care Research (NIHR) Health Protection Research Unit (HPRU) in Modelling and Health Economics (grant number NIHR200908) and the European Union's Horizon 2020 research and innovation programme - project EpiPose (101003688). The work of JB was supported by the Helmholtz Information and Data Science Project SIMCARD as well as Deutsche Forschungsgemeinschaft (DFG, German Research Foundation) – project number 512483310. SF's work was supported by the Wellcome Trust (grant: 210758/Z/18/Z) and the HPRU (grant code NIHR200908). The views expressed are those of the authors and not necessarily those of the UK Department of Health and Social Care (DHSC), NIHR, or UKHSA. The funders had no role in study design, data collection and analysis, decision to publish, or preparation of the manuscript.

**Competing interests:** The authors have declared that no competing interests exist.

## Author summary

Scores like the Continuous Ranked Probability Score (CRPS) or the Weighted Interval Score (WIS) are commonly used to evaluate epidemiological forecasts and are a measure of absolute distance between forecast and observation. Due to the exponential nature of epidemic processes, evaluating the absolute distance between forecast and observation may not be ideal. We argue that transforming counts before applying the CRPS or WIS can yield more meaningful results. The natural logarithm is a particularly attractive transformation in epidemiological settings. Scores computed on log-transformed values can be interpreted as a probabilistic version of a relative error and reflect how well forecasters predict the time-varying epidemic growth rate. If the data-generating process has a quadratic mean-variance relationship, the logarithmic transformation also leads to expected CRPS values which are independent of the order of magnitude of the predicted quantity. We illustrate these properties using data from the European COVID-19 Forecast Hub and find that scoring transformed counts changes model rankings. Stronger emphasis is given to situations in which forecasters missed the beginning of upward swings, while failing to predict a downturn following a peak is less severely penalised. We generally recommend including evaluations of transformed counts when assessing forecaster performance.

## Introduction

Probabilistic forecasts [1] play an important role in decision-making in epidemiology and public health [2], as well as other areas as diverse as economics [3] or meteorology [4]. Forecasts based on epidemiological modelling in particular have received widespread attention during the COVID-19 pandemic. Evaluations of forecasts can provide feedback for researchers to improve their models and train ensembles. They moreover help decision-makers distinguish good from bad predictions and choose forecasters and models that are best suited to inform future decisions.

Probabilistic forecasts are usually evaluated using so-called (strictly) proper scoring rules [5], which return a numerical score as a function of the forecast and the observed data. Proper scoring rules are constructed such that they encourage honest forecasting and cannot be 'gamed' or 'cheated'. Assuming that the forecaster's actual best judgement corresponds to a predictive distribution $F$, a proper score is constructed such that if $F$ was the data-generating process, no other distribution $G$ would yield a better expected score. A scoring rule is called *strictly* proper if there is no other distribution that under $F$ achieves the *same* expected score as $F$, meaning that any deviation from $F$ leads to a worsening of expected scores. Forecasters (anyone or anything that issues a forecast) are thus incentivised to report their true belief $F$ about the future. Common proper scoring rules are the logarithmic or log score [6] and the continuous ranked probability score (CRPS, [5]). The log score is the predictive log density or probability mass evaluated at the observed value. It is supported by the likelihood principle [7] and has many desirable theoretical properties; however, the particularly severe penalties it assigns to occasional misguided forecasts make it little robust [8]. Moreover, it is not easily applied to forecasts reported as samples or quantiles, as used in many recent disease forecasting efforts. It is nonetheless occasionally used in epidemiology (see e.g., [1, 9]), but in recent years the CRPS and the weighted interval score (WIS, [8]) have become increasingly popular.

The CRPS measures the distance of the predictive distribution to the observed data as

$$\text{CRPS}(F, y) = \int_{-\infty}^{\infty} \left( F(x) - \mathbf{1}(x \geq y) \right)^2 dx, \tag{1}$$

where $y$ is the true observed value, $F$ is the cumulative distribution function (CDF) of the predictive distribution, and $\mathbf{1}()$ is the indicator function. The CRPS can be understood as a generalisation of the absolute error to predictive distributions, and interpreted on the natural scale of the data. The WIS is an approximation of the CRPS for predictive distributions represented by a set of predictive quantiles and is currently used to assess forecasts in the so-called COVID-19 Forecast Hubs in the US [10, 11], Europe [12] and Germany and Poland [13, 14], as well as the US *FluSight* project on influenza forecasting [15]. The WIS is defined as

$$\text{WIS}(F, y) = \frac{1}{K} \times \sum_{k=1}^{K} 2 \times \left[ \mathbf{1}(y \leq q_{\tau_k}) - \tau_k \right] \times (q_{\tau_k} - y), \tag{2}$$

where $q_\tau$ is the $\tau$ quantile of the forecast $F$, $y$ is the observed outcome and $K$ is the number of (roughly equally spaced) predictive quantiles provided. The WIS can be decomposed into three components, dispersion, underprediction and overprediction, which reflect the spread of the forecast and whether it was centred above or below the observed value. We show an alternative definition based on central prediction intervals in S1 Text which illustrates this decomposition.

The notion of absolute distance encoded by the CRPS and WIS provides a straightforward interpretation, but may not always be the most useful perspective in the context of infectious disease spread. Especially in their early phase, outbreaks are best conceived as exponential processes, characterized by potentially time varying reproduction numbers $R_t$ [16] or epidemic growth rates $r_t$ [17]. If the true modelling task revolves around estimating and forecasting these quantities, then evaluating forecasts based on the absolute distance between forecasted and observed incidence values penalises underprediction (of the reproduction number or growth rate) less than overprediction by the same amount. For illustration, consider an incidence forecast issued at time 0 and referring to time $t$ that misses the correct average growth rate $\bar{r}_t$ by either $-\epsilon$ or $+\epsilon$. Then the ratio of the resulting absolute errors on the scale of observed incidences $y_t$ is

$$\frac{|y_0 \exp[(\bar{r}_t - \epsilon) \times t] - y_0 \exp(\bar{r}_t t)|}{|y_0 \exp[(\bar{r}_t + \epsilon) \times t] - y_0 \exp(\bar{r}_t t)|} = \exp(-\epsilon t) < 1. \tag{3}$$

If one is to measure the ability to forecast the underlying infection dynamics, it may thus be more desirable to evaluate errors on the scale of the growth rate directly.

Another argument against using notions of absolute distance between predicted and observed incidence values is that forecast consumers may find errors on a relative scale easier to interpret and more useful in order to track predictive performance across targets of different orders of magnitude. [18] have proposed the scaled CRPS (SCRPS) which is locally scale invariant; however, it does not correspond to a relative error measure and lacks a straightforward interpretation as available for the CRPS.

Lastly, it may be considered desirable to give all forecast targets similar weight in an overall performance evaluation. As the CRPS typically scales with the order of magnitude of the quantity to be predicted, this is not the case for the CRPS, which will typically assign higher scores to forecast targets with high expected values (e.g., in large locations or around the peak of an epidemic). Bracher et al. [8] have argued that this is a desirable feature, directing attention to situations of particular public health relevance. An evaluation based on absolute errors,

however, will assign little weight to other potentially important aspects, such as the ability to correctly predict future upswings while observed numbers are still low.

In many fields, it is common practice to forecast transformed quantities (see e.g. [19] in finance, [20] in macroeconomics, [21] in hydrology or [22] in meteorology). While the goal of the transformations is often to improve the accuracy of the predictions, they can also be used to enhance and complement the evaluation process. In this paper, we argue that the aforementioned issues with evaluating epidemic forecasts based on measures of absolute error on the natural scale can be addressed by transforming the forecasts and observations prior to scoring using some strictly monotonic transformation. Strictly monotonic transformations can shift the focus of the evaluation in a way that may be more appropriate for epidemiological forecasts, while guaranteeing that the score remains proper. Many different transformations may be appropriate and useful, depending on the exact context, the desired focus of the evaluation, and specific aspects forecast consumers care most about (see Discussion).

For conceptual clarity and to allow for a more in-depth discussion, we focus mostly on the natural logarithm as a particularly attractive transformation in the context of epidemic phenomena. We refer to this transformation as 'log-transformation' and to scores that have been computed from log-transformed forecasts and observations as scores 'on the log scale' (as opposed to scores 'on the natural scale', which involve no transformation). In the theoretical part of the paper, 'log-transformation' and 'log scale' generally refer to a transformation of $\log_e(x)$. For practical applications in the later sections we also use these terms to describe a transformation of $\log_e(x + a)$ with a small $a > 0$ in order to keep the terminology and notation simple. For a prediction target with strictly positive support, the CRPS after applying a log-transformation is given by

$$\text{CRPS}(F_{\log}, \log y) = \int_{-\infty}^{\infty} \left( F_{\log}(x) - \mathbf{1}(x \geq \log y) \right)^2 dx. \tag{4}$$

Here, $y$ is again the observed outcome and $F_{\log}$ is the predictive CDF of the log-transformed outcome, i.e.,

$$F_{\log}(x) = F(\exp(x)), \tag{5}$$

with $F$ the CDF on the original scale. Instead of a score representing the magnitude of absolute errors, applying a log-transformation prior to the CRPS yields a score which a) measures relative error, b) provides a measure for how well a forecast captures the exponential growth rate of the target quantity and c) is less dependent on the expected order of magnitude of the quantity to be predicted). We therefore argue that such evaluations on the logarithmic scale should complement the prevailing evaluations on the natural scale. Other transformations may likewise be of interest. We briefly explore the square root transformation as an alternative transformation. Our analysis mostly focuses on the CRPS (or WIS) as an evaluation metric for probabilistic forecasts, given its widespread use throughout the COVID-19 pandemic. We note that the logarithmic score has scale invariance properties which imply that score differences between different forecasts are invariant to strictly monotonic transformations (see [23] on corresponding properties of likelihood ratios and [24]). The question of the right scale to evaluate forecasts on does therefore not arise for the log score.

The remainder of the article is structured as follows. First, we provide some mathematical intuition on applying the log-transformation prior to evaluating the CRPS, highlighting the connections to relative error measures, the epidemic growth rate and variance stabilizing transformations. We then discuss the effect of the log-transformation on forecast rankings as well as practical considerations for applying transformations in general and the log-

transformation in particular. To analyse the real-world implications of the log-transformation we use forecasts submitted to the European COVID-19 Forecast Hub [12, 25]. Finally, we provide scoring recommendations, discuss alternative transformations that may be useful in different contexts, and suggest further research avenues).

## Logarithmic transformation of forecasts and observations

### Interpretation as a relative error

To illustrate the effect of applying the natural logarithm prior to evaluating forecasts we consider the absolute error, which the CRPS and WIS generalize to probabilistic forecasts. We assume strictly positive support (meaning that no specific handling of zero values is needed), a restriction we will address when applying this transformation in practice. When considering a point forecast $\hat{y}$ for a quantity of interest $y$, such that

$$y = \hat{y} + \varepsilon,\tag{6}$$

the absolute error is given by $|\varepsilon|$. When taking the logarithm of the forecast and the observation first, thus considering

$$\log y = \log \hat{y} + \varepsilon^*,\tag{7}$$

the resulting absolute error $|\varepsilon^*|$ can be interpreted as an approximation of various common relative error measures. Using that $\log(a) \approx a - 1$ if $a$ is close to 1, we get

$$|\varepsilon^*| = |\log \hat{y} - \log y| = \left|\log\left(\frac{\hat{y}}{y}\right)\right| \overset{\text{if } \hat{y} \approx y}{\approx} \left|\frac{\hat{y}}{y} - 1\right| = \left|\frac{\hat{y} - y}{y}\right|.\tag{8}$$

The absolute error after log transforming is thus an approximation of the *absolute percentage error* (APE, [26]) as long as forecast and observation are close. As we assumed that $\hat{y} \approx y$, we can also interpret it as an approximation of the *relative error* (RE, [26])

$$\left|\frac{\hat{y} - y}{\hat{y}}\right|\tag{9}$$

and the *symmetric absolute percentage error* (SAPE; see e.g., [27])

$$\left|\frac{\hat{y} - y}{y/2 + \hat{y}/2}\right|.\tag{10}$$

As Fig 1 shows, the alignment with the SAPE is in fact the closest and holds quite well even if predicted and observed value differ by a factor of two or three. Generalising to probabilistic forecasts, the CRPS applied to log-transformed forecasts and outcomes can thus be seen as a probabilistic counterpart to the symmetric absolute percentage error, which offers an appealing intuitive interpretation.

### Interpretation as scoring the exponential growth rate

Another interpretation for the log-transform is possible if the generative process is framed as exponential with a time-varying growth rate $r(t)$ (see e.g. [28]), i.e.

$$\frac{d}{dt}y(t) = r(t)y(t)\tag{11}$$

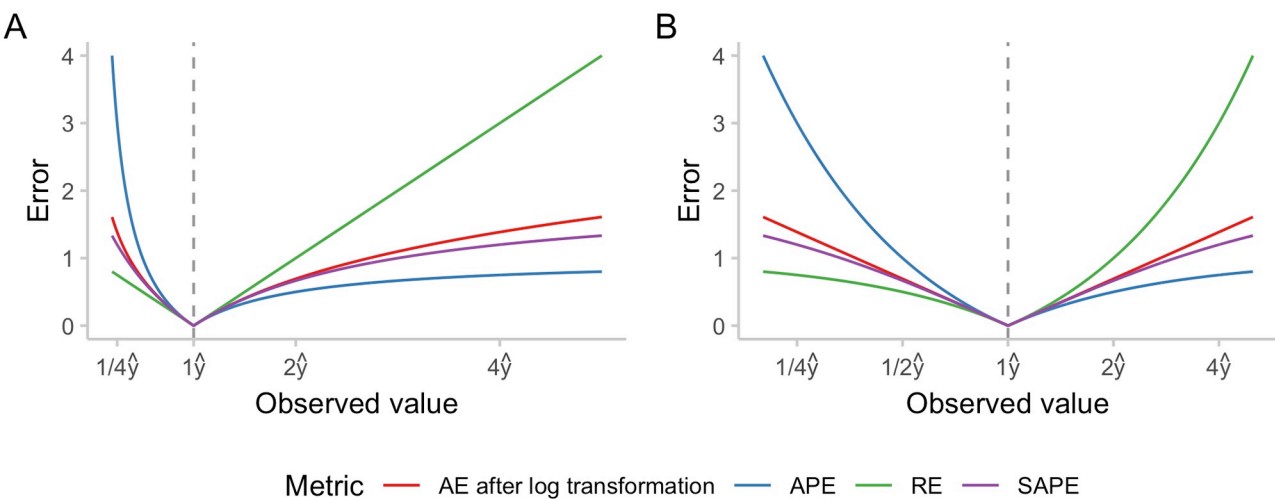

**Fig 1. Numerical comparison of different measures of relative error: Absolute percentage error (APE), relative error (RE), symmetric absolute percentage error (SAPE) and the absolute error applied to log-transformed predictions and observations.** We denote the predicted value by $\hat{y}$ and display errors as a function of the ratio of observed and predicted value. A: x-axis shown on a linear scale. B: x-axis shown on a logarithmic scale.

which is solved by

$$y(t) = y_0 \exp\left( \int_0^t r(t')dt' \right) = y_0 \exp(\bar{r}_t t) \qquad (12)$$

where $y_0$ is an initial data point and $\bar{r}_t$ is the mean of the growth rate between the initial time point 0 and time $t$.

If a forecast $\hat{y}(t)$ for the value of the time series at time $t$ is issued at time 0 based on the data point $y_0$ then the absolute error after log transformation is

$$
\begin{aligned}
\epsilon^* &= |\log\left[\hat{y}(t)\right] - \log\left[y(t)\right]| \\
&= |\log\left[y_0 \exp(\hat{\bar{r}}_t t)\right] - \log\left[y_0 \exp(\bar{r}_t t)\right]| \qquad (13) \\
&= t|\hat{\bar{r}}_t - \bar{r}_t|
\end{aligned}
$$

where $\bar{r}_t$ is the true mean growth rate and $\hat{\bar{r}}_t$ is the forecast mean growth rate. We thus evaluate the error in the mean exponential growth rate, scaled by the length of the time period considered. Again generalising this to the CRPS and WIS implies a probabilistic evaluation of forecasts of the epidemic growth rate.

## Interpretation as a variance-stabilising transformation

When evaluating models across sets of forecasting tasks, it may be desirable for each target to have a similar impact on the overall results. This could be motivated by the assumption that forecasts from different geographical units and time periods provide similar amounts of information about how well a forecaster performs. One would then like the resulting scores to be independent of the order of magnitude of the target to predict. CRPS values on the natural scale, however, typically scale with the order of magnitude of the quantity to be predicted. Average scores are then dominated by the results achieved for targets with high expected outcomes in a way that does not necessarily reflect the underlying predictive ability well.

If the predictive distribution for the quantity $Y$ equals the true data-generating process $F$ (an ideal forecast), the expected CRPS is given by [5]

$$\mathbb{E}[\text{CRPS}(F, y)] = 0.5 \times \mathbb{E}|Y - Y'|, \tag{14}$$

where $Y$ and $Y'$ are independent samples from $F$. This corresponds to half the *mean absolute difference*, which is a measure of dispersion. If $F$ is well-approximated by a normal distribution $\text{N}(\mu, \sigma^2)$, the approximation

$$\mathbb{E}_F[\text{CRPS}(F, y)] \approx \frac{\sigma}{\sqrt{\pi}} \tag{15}$$

can be used. This means that the expected CRPS scales roughly with the standard deviation, which in turn typically increases with the mean in epidemiological forecasting. In order to make the expected CRPS independent of the expected outcome, a *variance-stabilising transformation* (VST, [29, 30]) can be employed. The choice of this transformation depends on the mean-variance relationship of the underlying process.

If the mean-variance relationship of the data-generating distribution is quadratic with $\sigma^2 = c \times \mu^2$, the natural logarithm can serve as the VST. Denoting by $F_{\log}$ the predictive distribution for $\log(Y)$, we can use the delta method (a first-order Taylor approximation, see e.g., [30]), to show that

$$\mathbb{E}_F[\text{CRPS}\{F_{\log}, \log(y)\}] \approx \frac{\sigma/\mu}{\sqrt{\pi}} = \frac{\sqrt{c}}{\sqrt{\pi}}. \tag{16}$$

As $\sigma$ and $\mu$ are linked through the quadratic mean-variance relationship (or linear mean-standard deviation relationship, $\sigma = \sqrt{c} \times \mu$), the expected CRPS thus stays constant regardless of the expected value of the data-generating distribution $\mu$. The assumption of a quadratic mean-variance relationship is closely linked to the aspects discussed earlier. It implies that relative errors have constant variance and can thus be meaningfully compared across different targets. Also, it arises naturally if we assume that our capacity to predict the epidemic growth rate does not depend on the expected outcome, i.e. does not depend on the current phase of the epidemic or the order of magnitude of current observations.

If the mean-variance relationship is linear with $\sigma^2 = c \times \mu$, as with a Poisson-distributed variable, the square root is known to be a VST [30]. Denoting by $F_{\sqrt{\ }}$ the predictive distribution for $\sqrt{Y}$, the delta method can again be used to show that

$$\mathbb{E}_F[\text{CRPS}\{F_{\sqrt{\ }}, \sqrt{y}\}] \approx \frac{\sigma/\sqrt{\mu}}{2\sqrt{\pi}} = \frac{\sqrt{c}}{2\sqrt{\pi}}. \tag{17}$$

We note that while standard in the derivation of variance-stabilizing transformations, the application of the delta method in Eqs (16) and (17) requires the probability mass of $F$ to be tightly distributed. If this is not the case, the approximation and thus the variance stabilization may be less accurate.

To strengthen our intuition on how transforming outcomes prior to applying the CRPS shifts the emphasis between targets with high and low expected outcomes, Fig 2 shows the expected CRPS of ideal forecasters under different mean-variance relationships and transformations. We consider a Poisson distribution where $\sigma^2 = \mu$, a negative binomial distribution with size parameter $\theta = 10$ and thus $\sigma^2 = \mu + \mu^2/10$, and a truncated normal distribution with practically constant variance. We see that when applying the CRPS on the natural scale, the expected CRPS grows monotonically as the variance of the predictive distribution (which is equal to the data-generating distribution for the ideal forecaster) increases. The expected

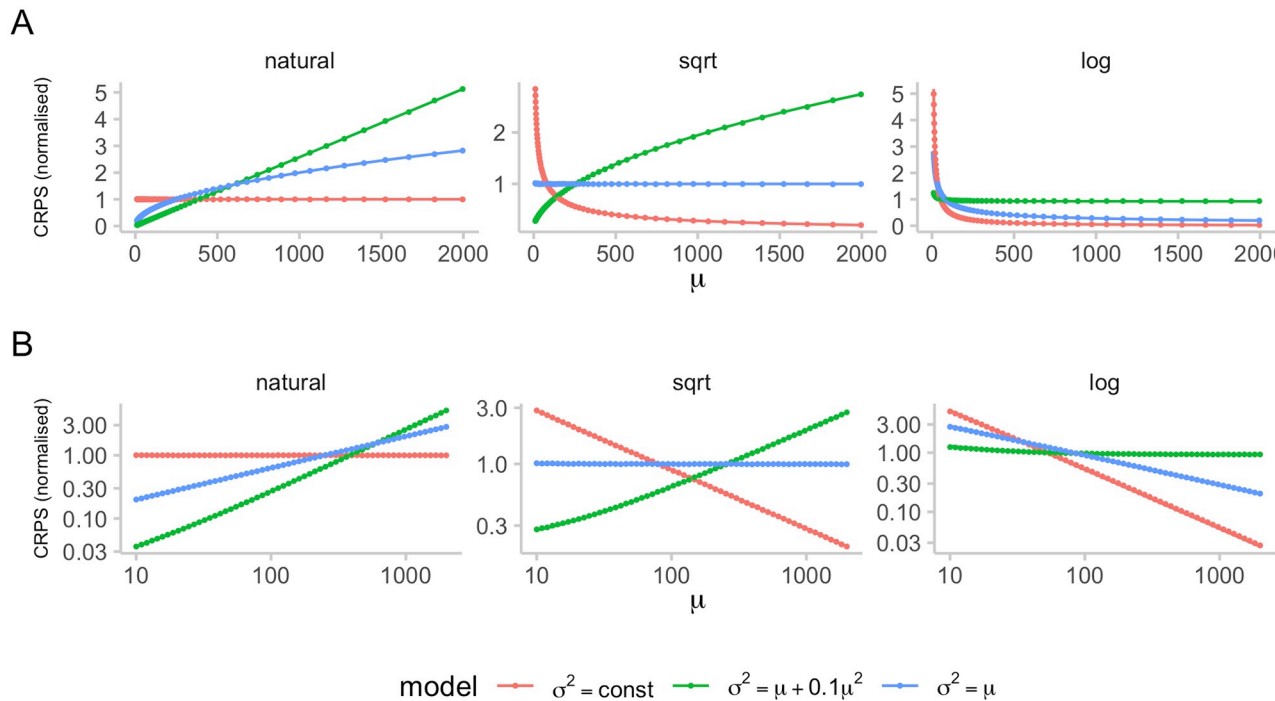

**Fig 2. Expected CRPS scores as a function of the mean and variance of the forecast quantity.** We computed expected CRPS values for three different distributions, assuming an ideal forecaster with predictive distribution equal to the true underlying (data-generating) distribution. These expected CRPS values were computed for different predictive means based on 10,000 samples each and are represented by dots. Solid lines show the corresponding approximations of the expected CRPS from Eqs (16) and (17). S3 Fig shows the quality of the approximation in more detail. The first distribution (red) is a truncated normal distribution with constant variance (we chose $\sigma = 1$ in order to only obtain positive samples). The second (green) is a negative binomial distribution with variance $\theta = 10$ and variance $\sigma^2 = \mu + 0.1\mu^2$. The third (blue) is a Poisson distribution with $\sigma^2 = \mu$. To make the scores for the different distributions comparable, scores were normalised to one, meaning that the mean score for every distribution (red, green, blue) is one. A: Normalised expected CRPS for ideal forecasts with increasing means for three distribution with different relationships between mean and variance. Expected CRPS was computed on the natural scale (left), after applying a square-root transformation (middle), and after adding one and applying a log-transformation to the data (right). B: A but with x and y axes on the log scale.

CRPS is constant only for the distribution with constant variance, and grows in $\mu$ for the other two. When applying a log-transformation first, the expected CRPS is almost independent of $\mu$ for the negative binomial distribution and large $\mu$, while smaller targets have higher expected CRPS in case of the Poisson distribution and the normal distribution with constant variance. When applying a square-root-transformation, the expected CRPS is independent of the mean for the Poisson-distribution, but not for the other two (with a positive relationship in the normal case and a negative one for the negative binomial). As can be seen in Fig 2 and S3 Fig, the approximations presented in Eqs (16) and (17) work quite well for our simulated example.

## Effects on model rankings

Rankings between different forecasters based on the CRPS may change when making use of a transformation, both in terms of aggregate and individual scores. We illustrate this in Fig 3 with two forecasters, A and B, issuing two different distributions with different dispersion. When showing the obtained CRPS as a function of the observed value, it can be seen that the ranking between the two forecasters may change when scoring the forecast on the logarithmic, rather than the natural scale. In particular, on the natural scale, forecaster A, who issues a more uncertain distribution, receives a better score than forecaster B for observed values far away from the centre of the respective predictive distribution. On the log scale, however,

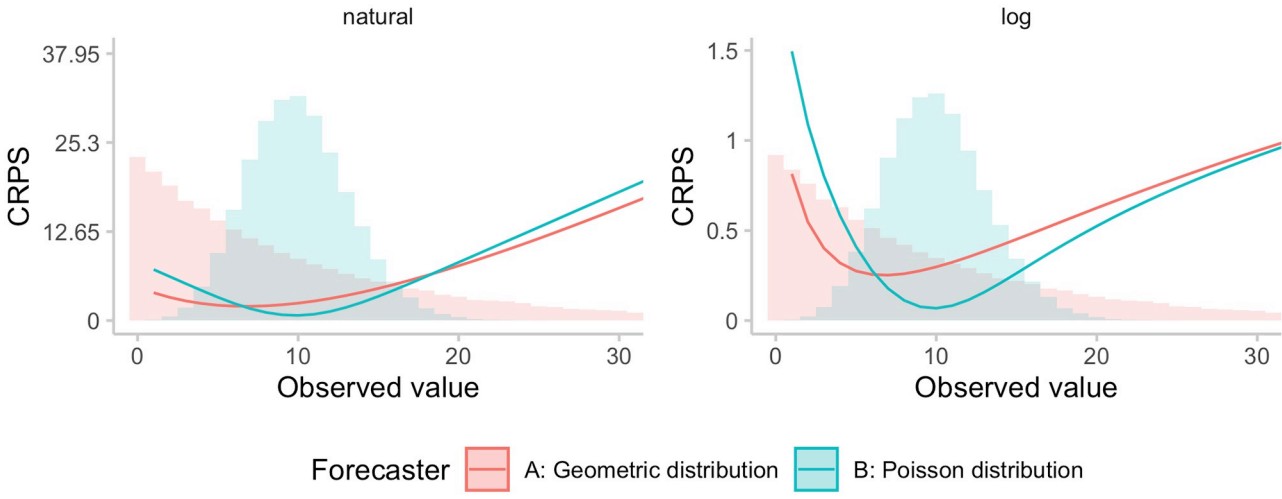

**Fig 3. Illustration of the effect of the log-transformation of the ranking for a single forecast.** Shown are CRPS (or WIS, respectively) values as a function of the observed value for two forecasters. Model A issues a geometric distribution (a negative binomial distribution with size parameter $\theta = 1$) with mean $\mu = 10$ and variance $\sigma^2 = \mu + \mu^2 = 110$), while Model B issues a Poisson distribution with mean and variance equal to 10. Zeroes in this illustrative example were handled by adding one before applying the natural logarithm.

forecaster A receives a lower score for large observed values, being more heavily penalised for assigning large probability to small values (which, in relative terms, are far away from the actual observation). We note that the chosen example involving a geometric forecast distribution is somewhat constructed; as illustrated in our practical example, rankings between models in practice stay quite stable for a single forecast.

Overall model rankings would be expected to differ more when scores are averaged across multiple forecasts or targets. The change in rankings of aggregate scores usually is mainly driven by the order of magnitude of scores for different forecast targets across time, location and target type and less so by the kind of changes in model rankings for single forecasts discussed above. Large observations will dominate average CRPS values when evaluation is done on the natural scale, but much less so after log transformation. Depending on how different models perform across targets of different orders of magnitude, rankings in terms of average scores may change when applying a transformation.

## Practical considerations and other transformations

In practice, one issue with the log transform is that it is not readily applicable to negative or zero values, which need to be removed or otherwise handled. One common approach to this end is to add a small positive quantity, such as $a = 1$, to all observations and predictions before taking the logarithm [31]. This still represents a strictly monotonic transformation, but the choice of $a$ does influence scores and rankings (measures of relative errors shrink the larger the chosen value $a$). As a rule of thumb, if if $x > 5a$, the difference between $\log(x + a)$ and $\log(x)$ is small, and it becomes negligible if $x > 50a$. Choosing a suitable offset $a$ thus balances two competing concerns: on the one hand, choosing a small $a$ makes sure that the transformation is as close to a natural logarithm as possible and scores can be interpreted as outlined in the previous sections. On the other hand, choosing a larger $a$ can help stabilise scores for forecasts and observations close to zero, avoiding giving excessive weight to forecasts of small quantities. For increasing $a$, less relative weight is given to smaller forecast targets. For very large values of $a$, $\log(x + a)$ is roughly linear in $x$, so that using a very large $a$ implies similar relative weighting as applying no transformation at all. In practice, a user could explore the effect of different

values of $a$ graphically and choose $a$ such that the relative weightings of times and regions with high and low incidence correspond to their preferences.

A related issue occurs when the predictive distribution has a large probability mass on zero (or on very small values), as this can translate into an excessively wide forecast in relative terms. In our applied example this is illustrated in S7 Fig. In such instances, the dispersion component of the WIS is inflated for scores obtained after applying the natural logarithm because forecasts contained zero in its prediction intervals. To deal with this issue one could choose to use a higher $a$ value when applying a transformation $\log(x + a)$, for example $a = 10$ instead of the $a = 1$ that we chose to use.

A natural question is which other transformations could be applied and whether resulting scores remain (strictly) proper. In principle, any transformation function can be applied simultaneously to forecasts and observations as long as the definition of the transformation is independent of the forecasts and any quantities unknown at the time of forecasting, including the observed value. This simply corresponds to a re-definition of the forecasting target. However, applying non-invertible transformations leads to a loss in information conveyed by forecasts, which we consider undesirable. The resulting score will be proper, but it may not be strictly proper anymore (as forecasts differing from the forecaster's true belief on the original scale may be identical on the transformed scale). When using the CRPS or the WIS, it seems most appropriate to use only strictly monotonic transformations such as the natural logarithm or the square root as otherwise the encoded notion of distance may become meaningless.

Some other strictly monotonic transformations that can be applied are scaling by the population size or scaling by past observations. The latter is similar to applying a log-transformation, but corresponds to evaluating a forecast of multiplicative, rather than exponential growth rates. The arising issue of dividing by zero can again be solved by adding a small offset $a$. Scaling a forecast by the later observed value (as opposed to scaling by past observations) is generally not permissible as it can result in improper scores (see [32] on the closely related topic of weighting scores with a function of the observed value). Similarly, scaling forecasts and observations by a function of the predictive distribution (like the predictive mean) may lead to improper scores; however, we are unaware of existing theoretical arguments on this.

When applying a transformation, the order of the operations matters, and applying a transformation after scores have been computed generally does not guarantee that the score remains proper. In the case of log transforms, taking the logarithm of the CRPS values, rather than scoring the log-transformed forecasts and data, results in an improper score. We illustrate this point using simulated data in S1 Fig, where it can be seen that in the example overconfident models perform best in terms of the log WIS. We note that strictly speaking, re-scaling average scores by the average score of a baseline model or average scores across different models to obtain skill scores likewise leads to improper scores [5]. The application of such skill scores, however, is established practice and considered largely unproblematic.

We note that in the practical evaluation of operational forecasting systems several additional challenges arise, which we do not study in detail. These concern e.g., the removal of outlying observations and forecasts and the handling of missing forecasts. The solutions we employed in practice are detailed below.

## Empirical example: The European Forecast Hub

### Setting

As an empirical comparison of evaluating forecasts on the natural and on the log scale, we use forecasts from the European Forecast Hub [12, 25]. The European COVID-19 Forecast Hub is one of several COVID-19 Forecast Hubs [11, 13] which have been systematically collecting,

aggregating and evaluating forecasts of several COVID-19 targets created by different teams every week. Forecasts are made one to four weeks ahead into the future and follow a quantile-based format with a set of 23 quantiles (0.01, 0.025, 0.05, . . ., 0.5, . . .0.95, 0.975, 0.99).

The forecasts used for the purpose of this illustration are forecasts submitted between the 8th of March 2021 and the 5th of December 2022 for reported cases and deaths from COVID-19. Target dates range from the 13th of March 2021 to the 10th of December 2022, for a total of 92 weeks. See [12] for a more thorough description of the data. We filtered all forecasts submitted to the Hub to only include the seven models which have submitted forecasts for both deaths and cases for 4 horizons in 32 locations on at least 46 forecast dates (see S4 Fig). We removed all observations marked as data anomalies by the European Forecast Hub [12] as well as all remaining negative observed values. These anomalies made up a relevant fraction of all observations. On average across locations, 12.1 out of 92 (13.2%) observations were removed for cases and 12.4 out of 92 (13.5%) for deaths. S5 Fig displays the number of anomalies removed for each location. In addition, we filtered out a small number of erroneous forecasts that were in extremely poor agreement with the observed data, as defined by any of the conditions listed in S2 Table. S6 Fig shows the percentage of forecasts removed for each model. Those few (less than 0.2% of forecasts for each model) erroneous outlier forecasts had excessive influence on average scores and relative skill scores in a way that was not representative of normal model behaviour. We removed them here in order to better illustrate the effects of the log-transformation on scores that one would expect in a well-behaved scenario. In a regular forecast evaluation such erroneous forecasts should usually not be removed and would count towards overall model scores.

All predictive quantiles were truncated at 0. We applied the log-transformation after adding a constant $a = 1$ to all predictions and observed values. The choice of $a = 1$ in part reflects convention, but also represents a suitable choice as it avoids giving excessive weight to forecasts close to zero, while at the same time ensuring that scores for observations >5 can be interpreted reasonably. S2 Fig illustrates the effect of adding a small quantity before taking the logarithm. The analysis was conducted in R [33], using the scoringutils package [34] for forecast evaluation. All code is available on GitHub (https://github.com/epiforecasts/transformation-forecast-evaluation). Where not otherwise stated, we report results for a two-week-ahead forecast horizon.

In addition to the WIS we use pairwise comparisons [11] to evaluate the relative performance of models across countries in the presence of missing forecasts. In the first step, score ratios are computed for all pairs of models by taking the set of overlapping forecasts between the two models and dividing the score of one model by the score achieved by the other model. The relative skill for a given model compared to others is then obtained by taking the geometric mean of all score ratios which involve that model. Low values are better, and the "average" model receives a relative skill score of 1.

## Illustration and qualitative observations

When comparing examples of forecasts on the natural scale with those on the log scale (see Fig 4, S7 and S8 Figs) a few interesting patterns emerge. Missing the peak, i.e. predicting increasing numbers while actual observations are already falling, tends to contribute a lot to overall scores on the natural scale (see forecasts during the peak in May 2022 in Fig 4A and 4B). On the log scale, these have less of an influence, as errors are smaller in relative terms (see Fig 4C and 4D). Conversely, failure to predict an upswing while numbers are still low, is less severely penalised on the natural scale (see forecasts in July 2021 and to a lesser extent in July 2022 in Fig 4A and 4B), as overall absolute errors are low. On the log scale, missing lower inflection

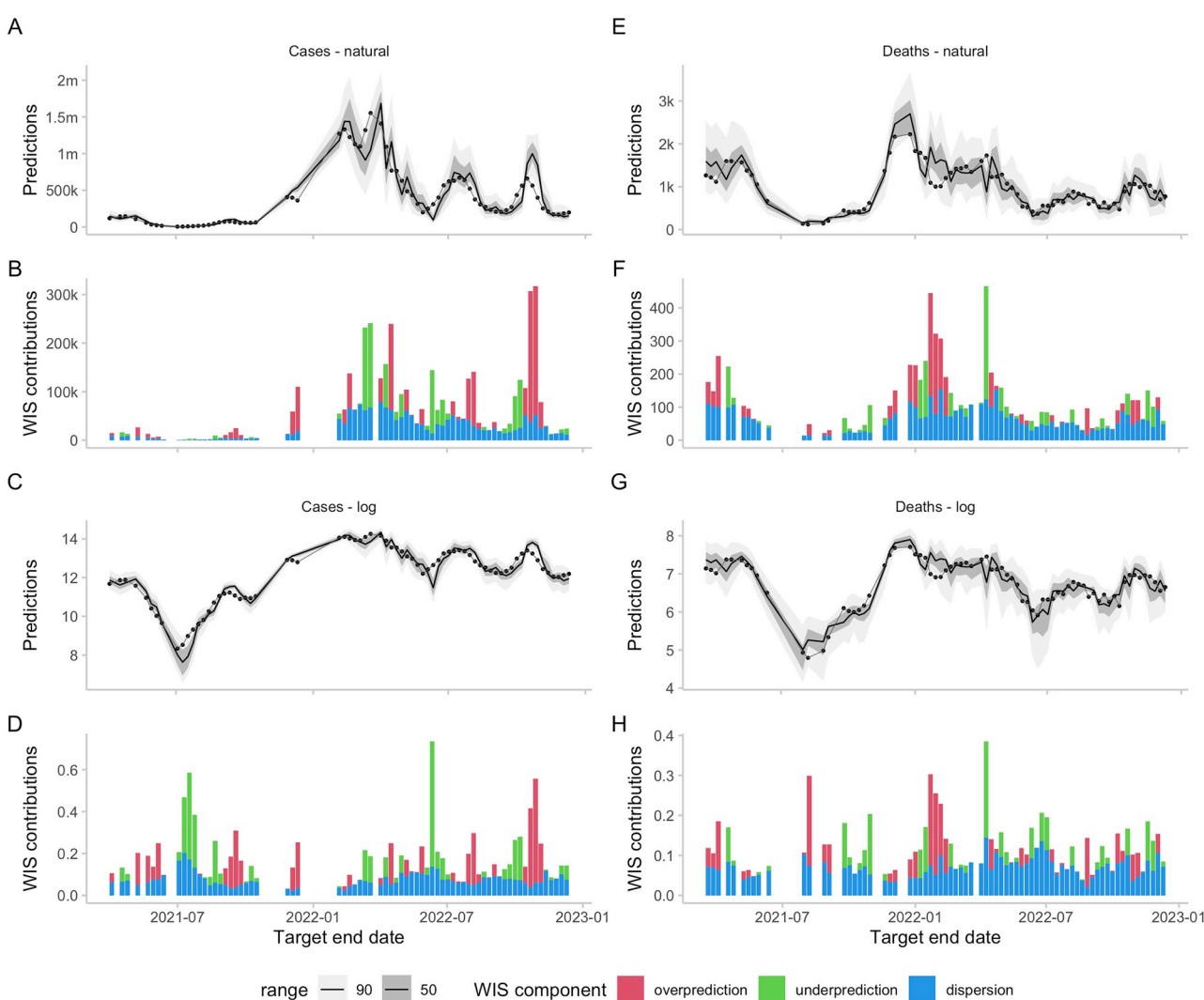

**Fig 4. Forecasts and scores for two-week-ahead predictions from the EuroCOVIDhub-ensemble made in Germany.** Missing values are due to data anomalies that were removed. A, E: 50% and 90% prediction intervals and observed values for cases and deaths on the natural scale. B, F: Corresponding scores. C, G: Forecasts and observations on the log scale. D, H: Corresponding scores.

points tends to lead to more severe penalties (see Fig 4C and 4D). One can also observe that on the natural scale, scores tend to track the overall level of the target quantity (compare for example forecasts for March-July with forecasts for September-October in Fig 4E and 4F). On the log scale, scores do not exhibit this behaviour and rather increase whenever forecasts are far away from the truth in relative terms, regardless of the overall level of observations.

Across the dataset, the average number of observed cases and deaths varied considerably by location and target type (see Fig 5A and 5B). On the natural scale, scores show a pattern quite similar to the observations across targets (see Fig 5D) and locations (see Fig 5C). On the log scale, scores were more evenly distributed between targets (see Fig 5D) and locations (see Fig 5C). Both on the natural scale as well on the log scale, scores increased considerably with increasing forecast horizon (see Fig 5E). This reflects the increasing difficulty of forecasts further into the future and, for the log scale, corresponds with our expectations based on the theoretical considerations detailed above.

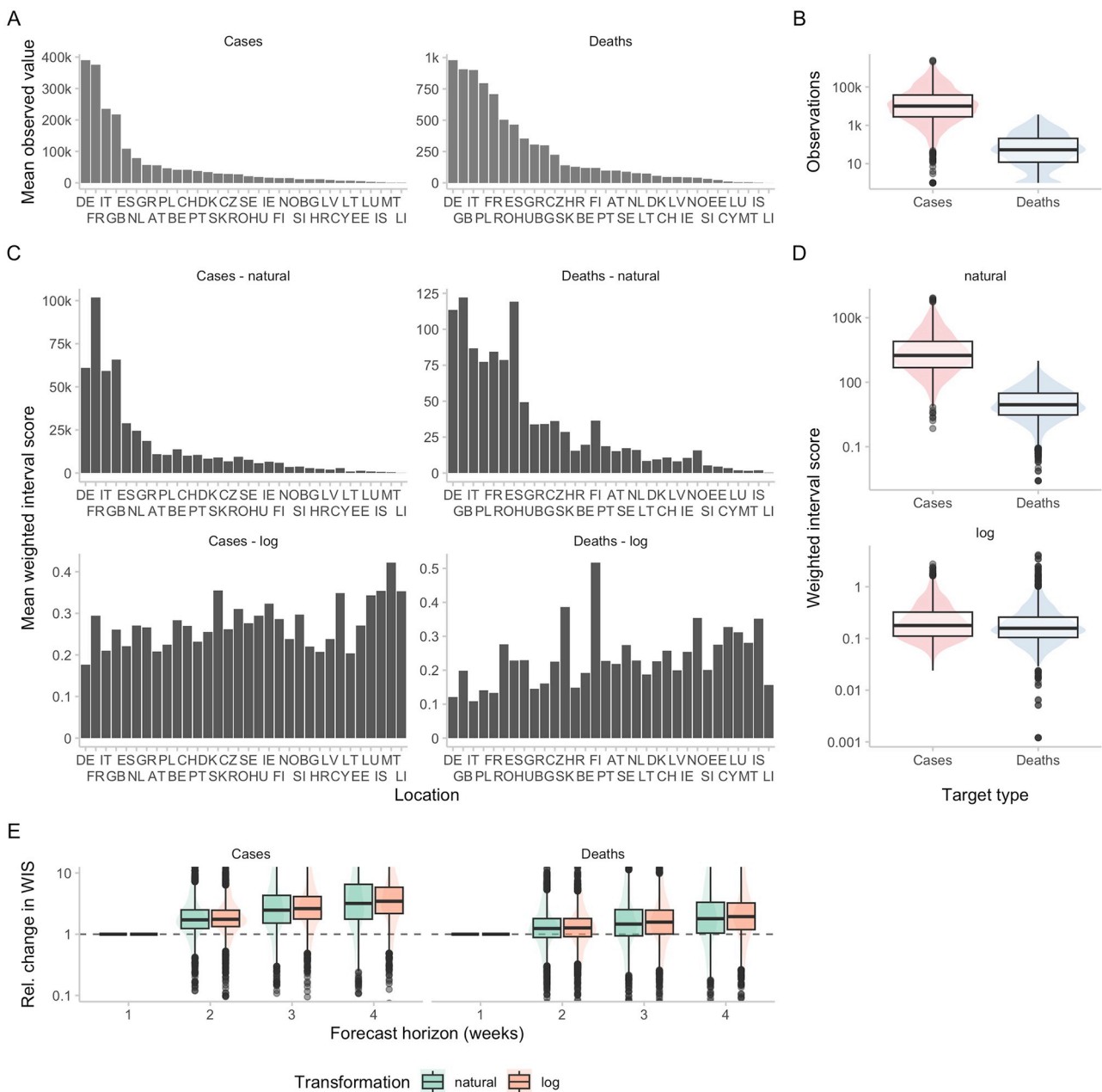

**Fig 5. Observations and scores across locations and forecast horizons for the European COVID-19 Forecast Hub data.** Locations are sorted according to the mean observed value in that location. A: Average (across all time points) of observed cases and deaths for different locations. B: Corresponding boxplot (y-axis on log-scale) of all cases and deaths. C: Scores for two-week-ahead forecasts from the EuroCOVIDhub-ensemble (averaged across all forecast dates) for different locations, evaluated on the natural scale as well as after transforming counts by adding one and applying the natural logarithm. D: Corresponding boxplots of all individual scores of the EuroCOVIDhub-ensemble for two-week-ahead predictions. E: Boxplots for the relative change of scores for the EuroCOVIDhub-ensemble across forecast horizons. For any given forecast date and location, forecasts were made for four different forecast horizons, resulting in four scores. All scores were divided by the score for forecast horizon one. To enhance interpretability, the range of visible relative changes in scores (relative to horizon = 1) was restricted to [0.1, 10].

To assess the impact of the choice of offset value *a* we extend the display from Fig 5C by results obtained under different specifications. Results are shown in Fig 6, where for completeness we also added the square root transformation. Smaller values of *a* increase the relative weight of smaller locations in the overall evaluation. In the most extreme considered

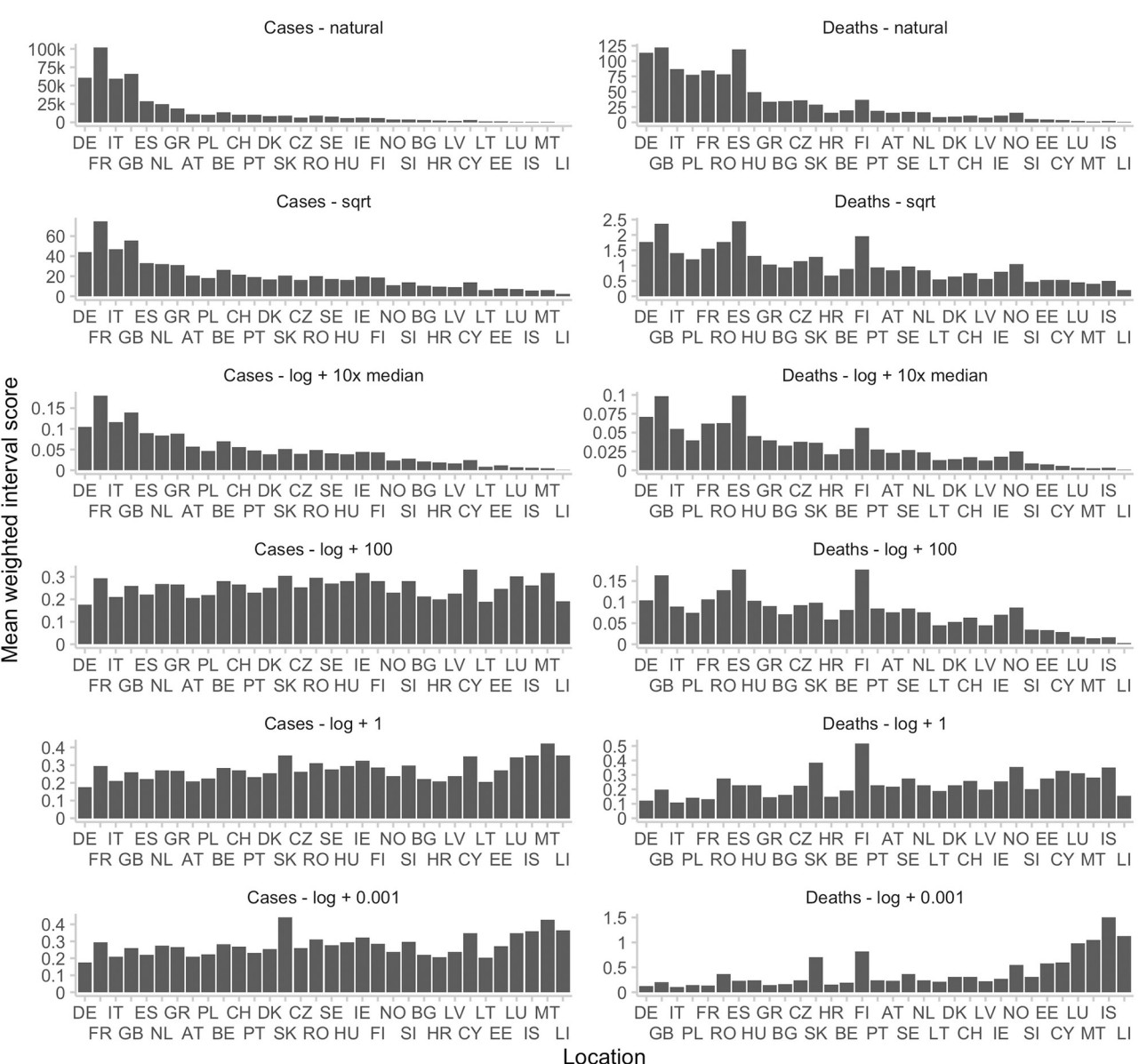

**Fig 6. Mean WIS in different locations for different transformations applied before scoring.** Locations are sorted according to the mean observed value in that location. Shown are scores for two-week-ahead forecasts of the EuroCOVIDhub-ensemble. On the natural scale (with no transformation prior to applying the WIS), scores correlate strongly with the average number of observed values in a given location. The same is true for scores obtained after applying a square-root transformation, or after applying a log-transformation with a large offset $a$. For illustrative purposes, $a$ was chosen to be 101630 for cases and 530 for deaths, 10 times the respective median observed value. For large values of $a$, $\log(x + a)$ grows roughly linearly in $x$, meaning that we expect to observe the same patterns as in the case with no transformation. For decreasing values of $a$, we give more relative weight to scores in small locations.

case $a = 0.001$, the smallest locations in fact receive the largest weight both for deaths and cases. For very large values (see the third row of Fig 6), the relative weights strongly resemble those of the evaluation on the natural scale. We recommend using displays of this type to get an intuition for the role different locations may play for overall evaluation results.

### Regression analysis to determine the variance-stabilizing transformation

As argued above, the mean-variance, or mean-CRPS, relationship determines which transformation can serve as a VST. We can analyse this relationship empirically by running a regression that explains the WIS (which approximates the CRPS) as a function of the central estimate of the predictive distribution. We ran the regression

$$\log[\text{WIS}(F, y)] = \alpha + \beta \times \log[\text{median}(F)], \tag{18}$$

where the predictive distribution $F$ and the observation $y$ are on the natural scale. This is equivalent to

$$\text{WIS}(F, y) = \exp(\alpha) \times \text{median}(F)^{\beta}, \tag{19}$$

meaning that we estimate a polynomial relationship between the predictive median and achieved WIS. Note that we are using predictive medians rather than means as only the former are available in the European COVID-19 Forecast Hub. As (under the simplifying assumption of normality; see the previous theoretical discussion on the mean-variance relationship) the WIS/CRPS of an ideal forecaster scales with the standard deviation, a value of $\beta = 1$ would imply a quadratic median-variance relationship; the natural logarithm could then serve as a VST. A value of $\beta = 0.5$ would imply a linear median-variance relationship, suggesting the square root as a VST. We applied the regression to case and death forecasts, stratified for one through four-week-ahead forecasts. Results are provided in Table 1. It can be seen that the estimates of $\beta$ always take a value somewhat below 1, implying a slightly sub-quadratic mean-variance relationship. The logarithmic transformation should thus approximately stabilize the variance (and WIS), possibly leading to somewhat higher scores for smaller forecast targets. The square-root transformation, on the other hand, can be expected to still lead to higher WIS values for targets of higher orders of magnitude.

To check the relationship after the transformation, we ran the regressions

$$\text{WIS}(F_{\log}, \log y) = \alpha_{\log} + \beta_{\log} \cdot \log(\text{median}(F)), \tag{20}$$

where $F_{\log}$ is the predictive distribution for $\log(y)$, and

$$\text{WIS}(F_{\sqrt{}}, \sqrt{y}) = \alpha_{\sqrt{}} + \beta_{\sqrt{}} \cdot \sqrt{\text{median}(F)}, \tag{21}$$

where $F_{\sqrt{}}$ is the predictive distribution on the square-root scale. A value of $\beta_{\log} = 0$ (or $\beta_{\sqrt{}} = 0$, respectively) would imply that scores are linearly independent of the median

**Table 1. Coefficients of three regressions for the effect of the magnitude of the median forecast on expected scores.** The first regression was $\log[\text{WIS}(F, y)] = \alpha + \beta \times \log[\text{median}(F)]$, where $F$ is the predictive distribution and $y$ the observed value. The second one was $\text{WIS}(F_{\log}, \log y) = \alpha_{\log} + \beta_{\log} \cdot \log(\text{median}(F))$, where $F_{\log}$ is the predictive distribution for $\log y$. The third one was $\text{WIS}(F_{\sqrt{}}, \sqrt{y}) = \alpha_{\sqrt{}} + \beta_{\sqrt{}} \cdot \sqrt{(\text{median}(F))}$, where $F_{\sqrt{}}$ is the predictive distribution for $\sqrt{y}$.

| Horizon | Target | $\alpha$ | $\beta$ | $\alpha_{\sqrt{}}$ | $\beta_{\sqrt{}}$ | $\alpha_{\log}$ | $\beta_{\log}$ |
|---------|--------|----------|---------|------------|------------|-----------|-----------|
| 1 | Cases | -0.862 | 0.876 | 0.790 | 0.087 | 0.433 | -0.024 |
| 2 | Cases | -0.243 | 0.877 | 0.959 | 0.162 | 0.660 | -0.031 |
| 3 | Cases | 0.372 | 0.855 | 1.109 | 0.238 | 0.882 | -0.037 |
| 4 | Cases | 0.816 | 0.837 | 1.645 | 0.296 | 1.009 | -0.036 |
| 1 | Deaths | -1.146 | 0.832 | 0.457 | 0.048 | 0.376 | -0.035 |
| 2 | Deaths | -0.981 | 0.867 | 0.443 | 0.084 | 0.416 | -0.028 |
| 3 | Deaths | -0.807 | 0.885 | 0.349 | 0.131 | 0.453 | -0.019 |
| 4 | Deaths | -0.602 | 0.891 | 0.125 | 0.194 | 0.501 | -0.011 |

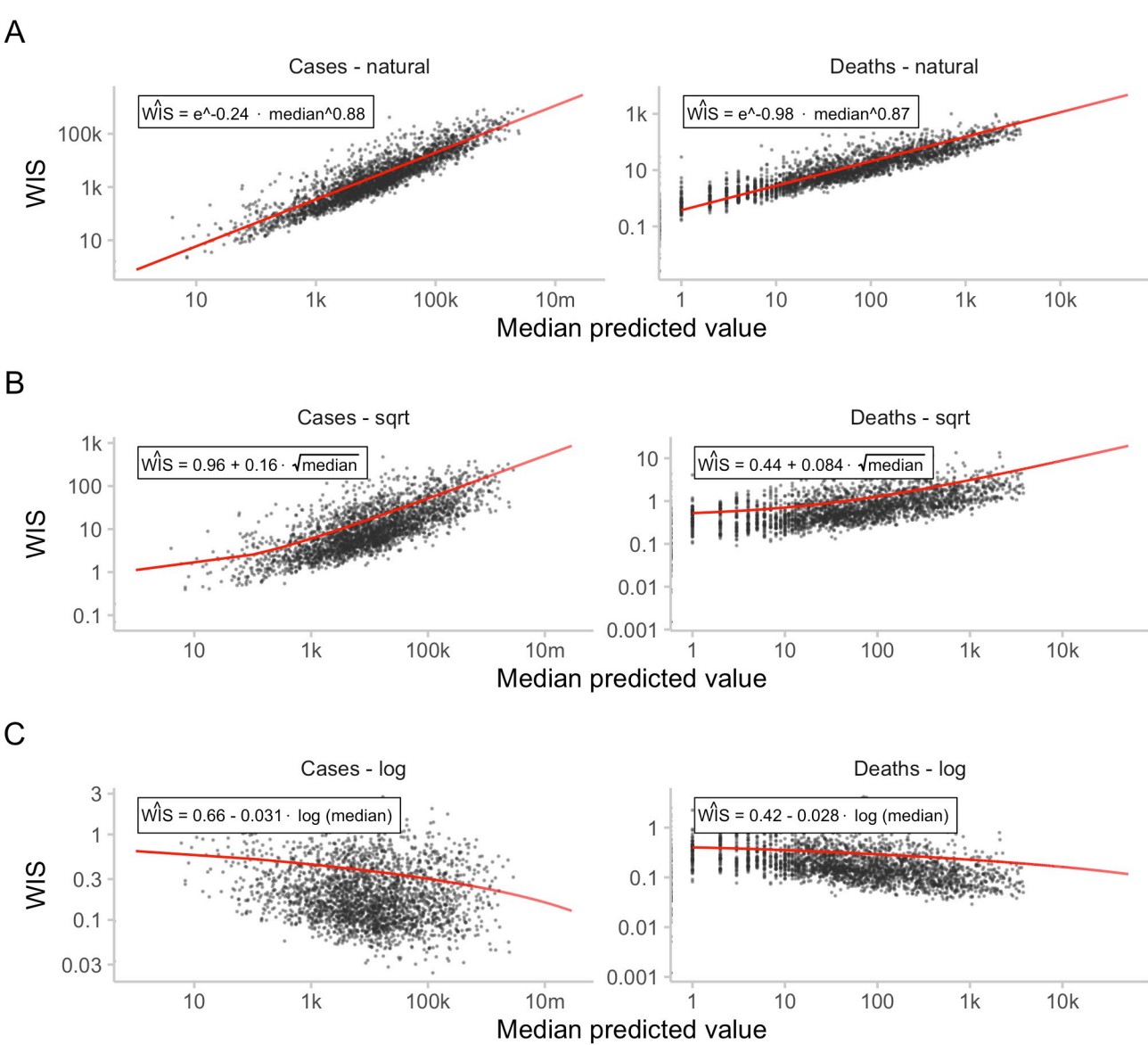

**Fig 7. Relationship between median forecasts and scores.** Black dots represent WIS values for two-week ahead predictions of the EuroCOVIDhub-ensemble. Drawn in red are the regression lines as discussed in the main text and shown in Table 1. A: WIS for two-week-ahead predictions of the EuroCOVIDhub-ensemble against median predicted values. B: Same as A, with scores obtained after applying a square-root-transformation to the data. C: Same as A, with scores obtained after applying a log-transformation to the data.

prediction after the transformation. A value smaller (larger) than 0 would imply that smaller (larger) targets lead to higher scores. As can be seen from Table 1, the results indeed indicate that small targets lead to larger average WIS when using the log transform ($\beta_{\log} < 0$), while the opposite is true for the square-root transform ($\beta_{\sqrt{\ }} > 0$). The results of the three regressions are also displayed in Fig 7. In this empirical example, the log transformation thus helps (albeit not perfectly), to stabilise WIS values, and it does so more successfully than the square-root transformation. As can be seen from Fig 7, the expected WIS scores for case targets with medians of 10 and 100,000 differ by more then a factor of ten for the square root transformation, but only a factor of around 2 for the logarithm.

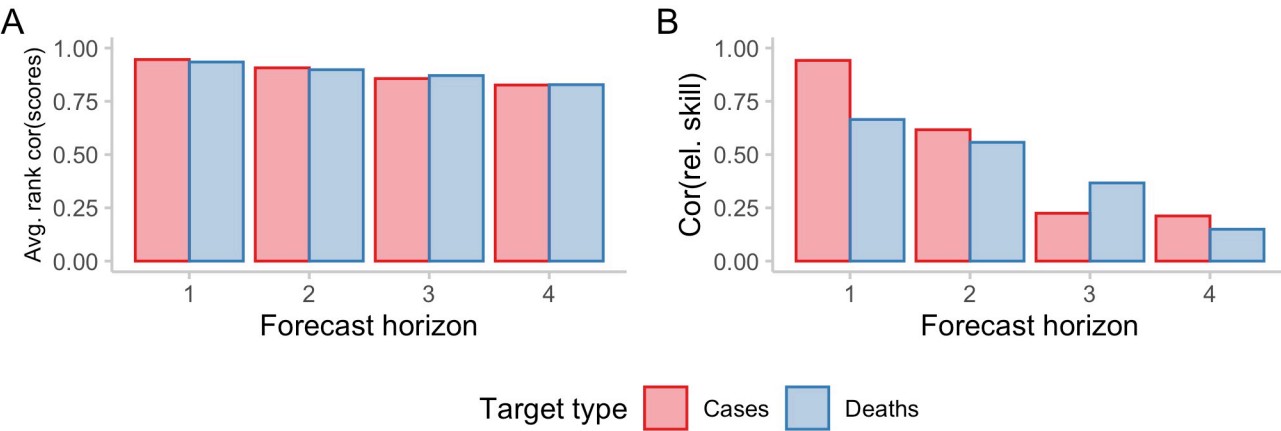

**Fig 8. Correlations of rankings on the natural and logarithmic scale.** A: Average Spearman rank correlation of scores for individual forecasts. For every individual target (defined by a combination of forecast date, target type, horizon, location), one score was obtained per model. Then, for every forecast target, the Spearman rank correlation was computed between scores on the natural scale and on the log scale for all the models that had made a forecast for that specific target. These individual rank correlations were then averaged across locations and time and are displayed stratified by horizon and target types, representing average accordance of model ranks for a single forecast target on the natural and on the log scale. B: Correlation between relative skill scores. For every forecast horizon and target type, a separate relative skill score was computed per model using pairwise comparisons, which is a measure of performance of a model relative to the others for a given horizon and target type that accounts for missing values. The plot shows the correlation between the relative skill scores on the natural vs. on the log scale, representing accordance of overall model performance as judged by scores on the natural and on the log scale.

### Impact of logarithmic transformation on model rankings

For *individual* forecasts, rankings between models for single forecasts are mostly preserved, with differences increasing across forecast horizons (see Fig 8A). While rankings between forecasters remain similar for a single forecast, this is not true anymore when looking at rankings obtained after averaging scores across multiple forecasts made at different times or in different locations. As discussed earlier, scores on the natural and on the log scale penalise errors very differently, e.g. when looking at performance during peaks or troughs. When evaluating performance *averaged across* different forecasts and forecast targets, relative skill scores of the models therefore change considerably (Fig 8B). The correlation between relative skill scores also decreases noticeably with increasing forecast horizon.

Fig 9 shows the changes in the ranking between different forecasting models. Encouragingly for the European Forecast Hub, the Hub ensemble, which is the forecast the organisers suggest forecast consumers make use of, remains the top model across scoring schemes. For cases, the ILM-EKF model and the Forecast Hub baseline model exhibit the largest change in relative skill scores. For the ILM-EKF model the relative proportion of the score that is due to overprediction is reduced when applying a log-transformation before scoring (see Fig 9E. Instances where the model has overshot are penalised less heavily on the log scale, leading to an overall better score. For the Forecast Hub baseline model, the fact that it often puts relevant probability mass on zero (see S7 Fig), leads to worse scores after applying log-transformation due to large dispersion penalties. For deaths, the baseline model seems to get similarly penalised for its in relative terms highly dispersed forecasts. The performance of other models changes as well, but patterns are less discernible on this aggregate level.

### Discussion

In this paper, we proposed the use of transformations, with a particular focus on the natural logarithmic transformation, when evaluating forecasts in an epidemiological setting. These

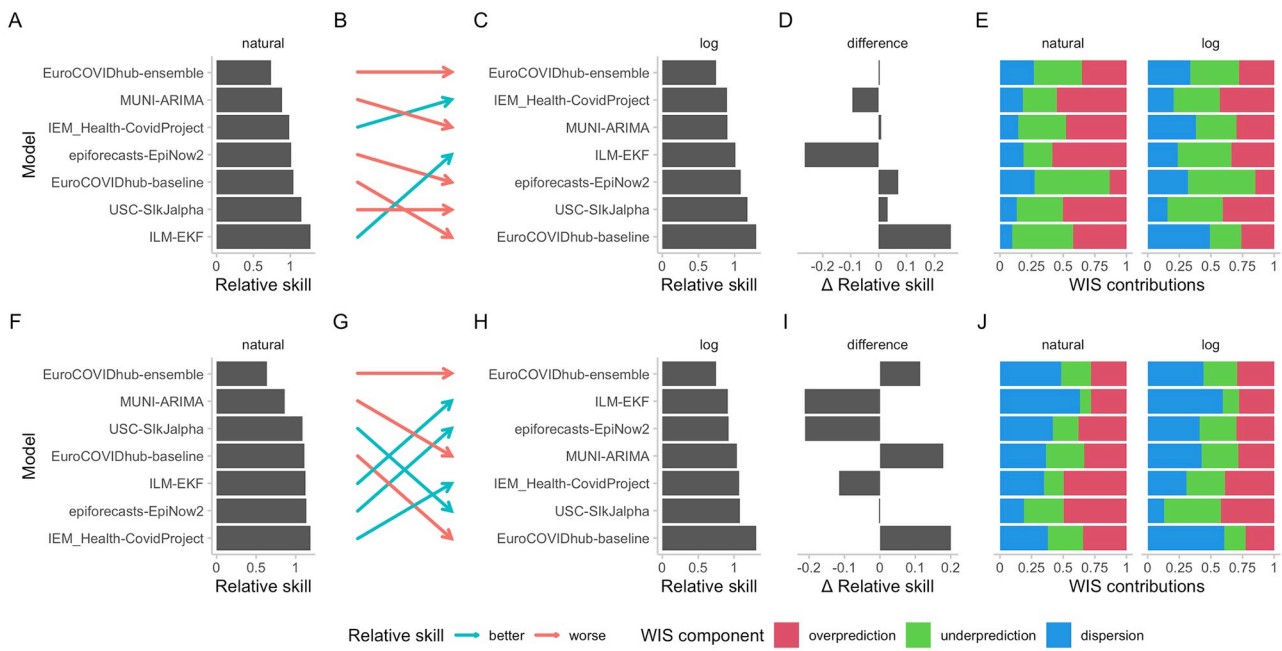

**Fig 9. Changes in model ratings as measured by relative skill for two-week-ahead predictions for cases (top row) and deaths (bottom row).** A: Relative skill scores for case forecasts from different models submitted to the European COVID-19 Forecast Hub computed on the natural scale. B: Change in rankings as determined by relative skill scores when moving from an evaluation on the natural scale to one on the logarithmic scale. Red arrows indicate that the relative skill scores deteriorated when moving from the natural to the log scale, green arrows indicate they improved. C: Relative skill scores based on scores on the log scale. D: Difference in relative skill scores computed on the natural and on the logarithmic scale, ordered as in C. E: Relative contributions of the different WIS components (overprediction, underprediction, and dispersion) to overall model scores on the natural and the logarithmic scale. F, G, H, I, J: Analogously for deaths.

transformations can address issues that arise when evaluating epidemiological forecasts based on measures of absolute error and their probabilistic generalisations (i.e CRPS and WIS). We showed that scores obtained after log-transforming both forecasts and observations can be interpreted as a) a measure of relative prediction errors, as well as b) a score for a forecast of the exponential growth rate of the target quantity and c) as variance stabilising transformation in some settings. When applying this approach to forecasts from the European COVID-19 Forecast Hub, we found overall scores on the log scale to be more equal across, time, location and target type (cases, deaths) than scores on the natural scale. Scores on the log scale were much less influenced by the overall incidence level in a country and showed a slight tendency to be higher in locations with very low incidences. We found that model rankings changed noticeably.

On the natural scale, missing the peak and overshooting was more severely penalised than missing the nadir and the following upswing in numbers. Both failure modes tended to be more equally penalised on the log scale (with undershooting receiving slightly higher penalties in our example).

Applying a log-transformation prior to the WIS means that forecasts are evaluated in terms of relative errors and errors on the exponential growth rate, rather than absolute errors. The most important strength of this approach is that the evaluation better accommodates the exponential nature of the epidemiological process and the types of errors forecasters who accurately model those processes are expected to make. The log-transformation also helps avoid issues with scores being strongly influenced by the order of magnitude of the forecast quantity, which can be an issue when evaluating forecasts on the natural scale. A potential downside is that forecast evaluation is unreliable in situations where observed values are zero or very small. One

could argue that this correctly reflect inherent uncertainty about the future course of an epidemic when numbers are small. Users nevertheless need to be aware that this can pose issues in practice. Including very small values in prediction intervals (see S7 Fig for an example) can lead to excessive dispersion values on the log scale. Similarly, locations with lower incidences may get disproportionate weight (i.e. high scores) when evaluating forecasts on the log scale. [8] argue that it is desirable to give large weight to forecasts for locations with high incidences, as this reflects performance on the targets we should care about most. On the other hand, scoring forecasts on the log scale may be less influenced by outliers and better reflect consistent performance across time, space, and forecast targets. Furthermore, decision makers may specifically care about situations in which numbers start to rise from a previously low level.

The log-transformation is only one of many transformations that may be useful and appropriate in an epidemiological context. One obvious option is to apply a population standardization to obtain incidence forecasts e.g., per 100,000 population [35]. We suggested using the natural logarithm as a variance-stabilising transformation (VST). This is appropriate for variables that are approximately normally distributed and have a quadratic mean-variance relationship with $\sigma^2 = c \times \mu^2$ (this is e.g. approximately true for the negative binomoial distribution and large $\mu$). Alternatively, the square-root transformation can be appropriate in the case of a Poisson distributed variable [30]. Other VST like the Box-Cox [36] are conceivable as well. If one is interested in multiplicative, rather than exponential growth rates, one could, instead of applying a log transformation, convert forecasts into forecasts for the multiplicative growth rate by dividing numbers by the last value that was observed at the time the forecast was made. Forecasters would then implicitly predict a separate multiplicative growth rate from today to horizon 1, 2, etc. Instead of dividing by the last observed value, another promising transformation would be to divide each forecast by the forecast of the previous week (and analogously for observations), in order to obtain forecasts for week-to-week growth rates. Alternatively, one could also take first differences of values on the log scale. This approach would be akin to evaluating the shape of the predicted trajectory against the shape of the observed trajectory (for a different approach to evaluating the shape of a forecast, see [37]). Dividing values by the previous value, unfortunately, is not feasible under the current quantile-based format of the Forecast Hubs, as the growth rate of the $\alpha$-quantile may be different from the $\alpha$-quantile of the growth-rate. However, it may be an interesting approach if predictive samples are available or if quantiles for weekwise growth rates have been collected. Potentially, the variance stabilising time-series forecasting literature may be a useful source of other transformations for various forecast settings.

It is possible to go beyond choosing a single transformation by constructing composite scores as a weighted sum of scores based on different transformations. This would make it possible to create custom scores and allow forecast consumers to choose and assign explicit weights to different qualities of the forecasts they might care about.

Exploring transformations is a promising avenue for future work that could help bridge the gap between modellers and policymakers by providing scoring rules that better reflect what forecast consumers care about. In this paper, we did not make any particular assumptions about policy makers' priorities and preferences. Rather, we aimed to enable users to make an informed choice by showing how different transformations lead to different relative weights for the kinds of prediction errors forecast consumers may care about, such as absolute vs. relative errors or the size of penalties for over- vs. underprediction. In practice, engagement with decision makers is important to determine what their priorities are and how different ways to measure predictive importance should be weighed.

We have shown that the natural logarithm transformation can lead to significant changes in the relative rankings of models against each other, with potentially important implications for decision-makers who rely on the knowledge of past performance to make a judgement about

which forecasts should inform future decisions. While it is commonly accepted that multiple proper scoring rules should usually be considered when comparing forecasts, we think this should be supplemented by considering different transformations of the data to obtain a richer picture of model performance. More work needs to be done to better understand the effects of applying transformations in different contexts, and how they may impact decision-making.

## Supporting information

**S1 Text. Alternative Formulation of the WIS.**
(PDF)

**S1 Table. Summary statistics for observations and scores for forecasts from the ECDC data set.**
(PDF)

**S2 Table. Criteria for removing forecasts.** Any forecast that met one of the listed criteria (represented by a row in the table), was removed. Those forecasts were removed in order to be better able to illustrate the effects of the log-transformation on scores and eliminating distortions caused by outlier forecasters. When evaluating models against each other (rather than illustrating the effect of a transformation), one would prefer not to condition on the outcome when deciding whether a forecast should be taken into account.
(PDF)

**S1 Fig. Illustration of the effect of applying a transformation after scoring.** We assume $Y \sim$ LogNormal(0, 1) and evaluate the expected CRPS for predictive distributions LogNormal(0, $\sigma$) with varying values of $\sigma \in [0.1, 2]$. For the regular CRPS (left) and CRPS applied to log-transformed outcomes (middle), the lowest expectation is achieved for the true value $\sigma = 1$. For the log-transformed CRPS, the optimal value is 0.9, i.e. there is an incentive to report a forecast that is too sharp. The score is therefore no longer proper.
(TIF)

**S2 Fig. Illustration of the effect of adding a small quantity to a value before taking the natural logarithm.** For increasing x, all lines eventually approach the black line (representing a transformation with no offset applied). For a given solid line, the dashed line of the same colour marks the x-value that is equal to 5 times the corresponding offset. It can be seen that for $a$ values smaller than one fifth of the transformed quantity, the effect of adding an offset is generally small. When choosing a suitable $a$, the trade-off is between staying close to the interpretation of a pure log-transformation (choosing a small $a$) and not giving excessive weights to small observations (by choosing a larger $a$, see Fig 6).
(TIF)

**S3 Fig. Visualisation of expected CRPS values against approximated scores.** This is using the approximation detailed in theoretical discussion on model rankings (see also Fig 2). Expected CRPS scores are shown for three different distributions once on the natural scale (top row) and once scored on the log scale (bottom row).
(TIF)

**S4 Fig. Number of forecasts available from different models for each forecast date.**
(TIF)

**S5 Fig. Number of observed values that were removed as anomalous.** The values were marked as anomalous by the European Forecast Hub team.
(TIF)

**S6 Fig. Number of forecasts marked as erroneous and removed.** Forecasts that were in extremely poor agreement with the observed values were removed from the analysis according to the criteria shown in S2 Table.
(TIF)

**S7 Fig. Forecasts and scores for two-week-ahead predictions from the EuroCOVIDhub-baseline made in Germany.** The model had zero included in some of its 50 percent intervals (e.g. for case forecasts in July 2021), leading to excessive dispersion values on the log scale. One could argue that including zero in the prediction intervals constituted an unreasonable forecast that was rightly penalised, but in general care has to be taken with small numbers. One potential way to do deal with this could be to use a higher $a$ value when applying a transformation $\log(x + a)$, for example $a = 10$ instead of $a = 1$. A, E: 50% and 90% prediction intervals and observed values for cases and deaths on the natural scale. B, F: Corresponding scores. C, G: Forecasts and observations on the log scale. D, H: Corresponding scores.
(TIF)

**S8 Fig. Forecasts and scores for two-week-ahead predictions from the epiforecasts-EpiNow2 model made in Germany.** A, E: 50% and 90% prediction intervals and observed values for cases and deaths on the natural scale from the EpiNow2 model [38]. B, F: Corresponding scores. C, G: Forecasts and observations on the log scale. D, H: Corresponding scores.
(TIF)

## Author Contributions

**Conceptualization:** Nikos I. Bosse, Sam Abbott, Anne Cori, Edwin van Leeuwen, Johannes Bracher, Sebastian Funk.

**Data curation:** Nikos I. Bosse.

**Formal analysis:** Nikos I. Bosse, Sam Abbott, Johannes Bracher, Sebastian Funk.

**Investigation:** Nikos I. Bosse, Sam Abbott, Johannes Bracher, Sebastian Funk.

**Methodology:** Nikos I. Bosse, Sam Abbott, Johannes Bracher, Sebastian Funk.

**Project administration:** Nikos I. Bosse.

**Resources:** Sebastian Funk.

**Software:** Nikos I. Bosse.

**Supervision:** Sam Abbott, Anne Cori, Edwin van Leeuwen, Johannes Bracher, Sebastian Funk.

**Validation:** Nikos I. Bosse, Sam Abbott, Johannes Bracher.

**Visualization:** Nikos I. Bosse, Johannes Bracher.

**Writing – original draft:** Nikos I. Bosse.

**Writing – review & editing:** Nikos I. Bosse, Sam Abbott, Anne Cori, Edwin van Leeuwen, Johannes Bracher, Sebastian Funk.

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
