## [Decision Letter · Decision Letter 0]

19 Apr 2023

Dear Mr Bosse,

Thank you very much for submitting your manuscript "Transformation of forecasts for evaluating predictive performance in an epidemiological context" for consideration at PLOS Computational Biology. As with all papers reviewed by the journal, your manuscript was reviewed by members of the editorial board and by several independent reviewers. The reviewers appreciated the attention to an important topic. Based on the reviews, we are likely to accept this manuscript for publication, providing that you modify the manuscript according to the review recommendations.

Sincerely,

James M McCaw, PhD

Academic Editor

PLOS Computational Biology

Thomas Leitner

Section Editor

PLOS Computational Biology

Reviewer's Responses to Questions

**Comments to the Authors:**

Reviewer #1: # Overview

Bosse et al. discuss the important problem of forecast evaluation in the context of epidemiological models. They make a clear and compelling argument for the use of different variance-stabilising transformations to reduce the impact of variation in model outputs and data over several orders of magnitude, a consequence of the exponential nature of epidemic growth (with time-varying growth rate).

I only really have minor, hopefully constructive points to raise — overall I think this is a very nice piece of work that deserves to be published in close to its current form. My big picture and lower level comments and suggestions follow.

# Big picture

## Proper scoring rules

As noted by the authors, proper scoring rules are a commonly used concept in evaluating probabilistic predictions. While probably familiar to a reasonable fraction of readers, I think there is also likely a reasonable set of readers who are not aware of these or are only vaguely aware of them. For this reason I think a little more general background would help. For example on line 38 after the ‘report their true belief about nature’ another sentence or two along the lines of

“For example, some methods of ‘scoring’ probabilistic predictions can be ‘gamed’ in the sense that forecasters can do better by reporting a probability distribution different to their best estimate. A proper scoring rule ensures that the expected score when reporting a distribution Q, as evaluated under their actual best estimate of the probability distribution P, is best when Q = P. Thus a proper scoring rule encourages a forecaster to provide ‘honest’ predictions."

In addition, I suspect there are a few readers (such as myself!) who are more familiar with more ‘basic’ scores such as e.g. logarithmic score or deviance (i.e. essentially the likelihood or related quantities). I spent a bit of time wondering about why these weren’t considered, especially as these avoid the scale issues at the centre of the present article. Such scores were only briefly considered in the discussion, in the second-to-last paragraph. I think it would be helpful to mention/emphasise up front some of these ‘classic’ scores that don’t have the same issues with scaling and then discuss why they aren’t used, e.g. due to ‘robustness’ or other issues (e.g. lack of full probability distributions from forecasts).

Minor changes such as these would, imo, help set the scene for the rest of the article a bit better.

## Log vs other transformations

The authors take the position that the log transformation is generally preferable for their use case, though they discuss and compare others. Relatedly, their arguments in terms of relative error also involve an essentially log-linear (i.e. log data + additive error) form which may be pragmatic but somewhat inelegant imo.

As the authors note, a particularly controversial issue for the log transformation is the issue of zero values. They take the standard pragmatic stance that we can use log (eps + y) instead of log y in such cases. Many communities e.g. econometrics are vehemently opposed to this, preferring e.g. quasi-Poisson regression and robust standard errors in the context of estimation. While not completely opposed to log (eps + y) myself (I have used it too!) it makes me a bit uncomfortable I still can’t help but feel that something along the lines of e.g. logarithmic/quasi-likelihood or deviance scores could be formulated that would be preferable to the log data transform. This would offer both automatic scaling and handling of zero values, in principle. However, I haven’t thought carefully enough to offer a concrete alternative and the log(eps + y) approach appears to work reasonably here. Instead, and in combination with the previous point, perhaps a bit more discussion of the potential alternatives based on quasi-likelihood-style functions rather than data transformations could be added?

## Observation and process models

The interpretation in e.g. 2.2 appears to essentially assume a deterministic process model and additive error on the log scale (multiplicative on the natural scale). Although more of a motivating heuristic than strict assumption, a deterministic process model based on a mean will not in general be the same as a stochastic process model, right? Perhaps a further caveat that this is a fairly simplistic motivating tool might be useful?

Furthermore, why not assume that the model mean (say the output of the deterministic model) defines e.g. the mean of something like a Poisson distribution (probably in overdispersed/quasi form)? or negative binomial? The potential use of these distributions is considered in later sections in the context of motivating variance transforms, but then would again seem to motivate a (quasi-)likelihood-style score beyond the approximate transforms (though with pros and cons in terms of robustness, applicability in the presence of partial information).

# More specific

I realised as I was about comment on some equations that none of the equations are numbered. It would probably be good to number them :-)

The first equation (line 43) uses an unbolded indicator function which isn’t defined (this also appears in the same form in the third equation, line 99). The second equation in contrast uses bold and defines the indicator. The definition should be moved forward and a choice of bold or not made.

I think the term ‘propriety’ (as in having the property of being proper) should either be explicitly defined or re-worded in terms of ‘being proper’.

No reference is provided for the approximate variance-stabilising properties of the square root transformation.

Reviewer #2: Please see attached comments.

Reviewer #3: In this manuscript the authors investigate how common scoring rules for probabilistic forecasts can be adjusted by transforming observed data and forecast predictions prior to scoring. They provide three clear motivations for applying such transformations prior to scoring, assess the implications of different transformations in great detail, and evaluate the effects of these transformations using real-world COVID-19 forecasts obtained from the European Forecast Hub. Approaches that can help us improve how we measure the utility of epidemic forecasts for decision-makers are definitely needed. This is a very nice and timely contribution to epidemic forecast evaluation, and I only have a few comments.

1. Line 96, typo: "log-transformtation".

2. Figure 3: out of curiosity, do the rankings of forecasts A and B first switch at the same observed value (somewhere around 7?) in both panels? Zooming in, it looks as though maybe the switch occurs very slightly earlier for the log-transformed scoring.

3. Section 3.2, page 9: remarks concerning Figure 4 refer to months (e.g., May, July) but it's a little difficult to relate them to the figure, because the x-axis only has 6-month breaks and spans two years, so it isn't immediately apparent which year (or both?) the reader should focus on. That being said, it's great to see how the log-transformation yields similar scores when the forecasts miss peaks and troughs.

4. Line 281: missing closing parenthesis, "(or \\Beta_\\sqrt = 0, respectively".

5. Figure 7: I can appreciate that the correlation between the natural and logarithmic scores decreases over the forecast horizon, as the absolute forecast error grows. But I found panel B more challenging to interpret. I gather the key message is that model rankings are quite consistent between the natural and log scales (panel A) even though the absolute scores differ markedly between the two scales — an effect that increases over the forecast horizon (panel B). Perhaps adding a sentence to this effect in the accompanying text (lines 292-295) would help other readers to avoid my confusion.

6. Line 296, typo: "Figure Figure 8 shows ...".

7. Section 3.4, page 14 and Figure 8: it's a welcome outcome that the Hub ensemble forecast remained the top model, given that the log-transformation clearly affects the individual model rankings. Presumably it would be reasonable to expect that ensemble forecasts for other regions, pathogens, etc, should be the best (or near-best) model when using log-transformed case counts?

8. Lines 327-329: "A potential downside is that forecast evaluation is unreliable in situations where observed values are zero or very small."

I feel there's an argument to be made here that rather than being a potential limitations, it could be considered an accurate reflection of the inherent uncertainty about the future course of an epidemic when case numbers are very small. This transformation also provides a valuable benefit in these situations, as the authors note a few sentences later: "It also gives higher weight to another type of situation one may care about, namely one in which numbers start to rise from a previously low level". This neatly illustrates the importance (as highlighted by the authors in this manuscript) of using multiple scoring rules to evaluate and compare forecasts.

9. Table SI.2, "Criteria for removing forecasts": forecasts were removed if their median prediction differed greatly from the true value. This should probably be mentioned in the manuscript text, somewhere around line 235 (where Table SI.2 is referenced), to notify the reader that the definition of "erroneous forecasts" includes forecasts that are in extremely poor agreement with the ground truth. Otherwise, I feel that "erroneous" may be open to misinterpretation (e.g., only removing forecasts that predicted negative counts, NaN values, etc).

**Have the authors made all data and (if applicable) computational code underlying the findings in their manuscript fully available?**

Reviewer #1: Yes

Reviewer #2: Yes

Reviewer #3: Yes

PLOS authors have the option to publish the peer review history of their article (what does this mean?). If published, this will include your full peer review and any attached files.

Reviewer #1: **Yes: **Oliver J. Maclaren

Reviewer #2: No

Reviewer #3: **Yes: **Robert Moss

Figure Files:

Data Requirements:

Reproducibility:

References:

---

## [Editor Report · Decision Letter 1]

27 Jul 2023

Dear Mr Bosse,

We are pleased to inform you that your manuscript 'Scoring epidemiological forecasts on transformed scales' has been provisionally accepted for publication in PLOS Computational Biology.

Best regards,

James M McCaw, PhD

Academic Editor

PLOS Computational Biology

Thomas Leitner

Section Editor

PLOS Computational Biology

---

## [Editor Report · Acceptance letter]

21 Aug 2023

PCOMPBIOL-D-23-00137R1 

Scoring epidemiological forecasts on transformed scales

Dear Dr Bosse,

I am pleased to inform you that your manuscript has been formally accepted for publication in PLOS Computational Biology. Your manuscript is now with our production department and you will be notified of the publication date in due course.

With kind regards,

Zsofi Zombor
